# Tyr198 is the Essential Autophosphorylation Site for STK16 Localization and Kinase Activity

**DOI:** 10.3390/ijms20194852

**Published:** 2019-09-30

**Authors:** Junjun Wang, Juanjuan Liu, Xinmiao Ji, Xin Zhang

**Affiliations:** 1High Magnetic Field Laboratory, Key Laboratory of High Magnetic Field and Ion Beam Physical Biology, Hefei Institutes of Physical Science, Chinese Academy of Sciences, Hefei 230031, China; wjunjun@mail.ustc.edu.cn (J.W.); xinmiaoji@hmfl.ac.cn (X.J.); 2Science Island Branch of Graduate School, University of Science and Technology of China, Hefei 230026, China; 3School of Life Sciences, Anhui University, Hefei 230601, China; 17067@ahu.edu.cn; 4Institutes of Physical Science and Information Technology, Anhui University, Hefei 230601, China

**Keywords:** STK16, kinase activity, auto-phosphorylation, Golgi apparatus, membrane, cell cycle

## Abstract

STK16, reported as a Golgi localized serine/threonine kinase, has been shown to participate in multiple cellular processes, including the TGF-β signaling pathway, TGN protein secretion and sorting, as well as cell cycle and Golgi assembly regulation. However, the mechanisms of the regulation of its kinase activity remain underexplored. It was known that STK16 is autophosphorylated at Thr185, Ser197, and Tyr198 of the activation segment in its kinase domain. We found that STK16 localizes to the cell membrane and the Golgi throughout the cell cycle, but mutations in the auto-phosphorylation sites not only alter its subcellular localization but also affect its kinase activity. In particular, the Tyr198 mutation alone significantly reduced the kinase activity of STK16, abolished its Golgi and membrane localization, and affected the cell cycle progression. This study demonstrates that a single site autophosphorylation of STK16 could affect its localization and function, which provides insights into the molecular regulatory mechanism of STK16’s kinase activity.

## 1. Introduction

STK16 (Ser/Thr Kinase 16, also known as Krct/PKL12/MPSK1/TSF-1) is a myristoylated and palmitoylated Ser/Thr protein kinase. It was firstly identified from mouse fetal liver (E12) in 1998 and was reported to be ubiquitously expressed and conserved among all eukaryotes [1,2,3,4,5]. Via interaction with various proteins and DNA substrates, STK16 is involved in the regulation of many different cellular processes, including TGF-β and VEGF transcription [5], GlcNAcK translocation [6], protein sorting of TGN and constitutive secretion [7,8], cell growth and cell differentiation [9]. As a membrane-associated kinase, it is primarily localized to the Golgi apparatus [10,11]. Moreover, in our recent studies, by using a highly selective small molecule inhibitor STK16-IN-1 and STK16 RNAi, in combination with *in vitro* actin polymerization and depolymerization assays, we found STK16 interacts with actin and regulates actin dynamics, which further affects Golgi assembly and cell cycle [10,12]. Although these studies demonstrate that STK16 is an important regulatory kinase, but how its own localization and function are regulated is still unknown.

For many protein kinases, posttranslational modifications, especially phosphorylation, are required and regulated tightly to avoid physiological disorder [13,14,15,16]. Among the whole structure, phosphorylation of amino acids in the activation loop is important for the proper orientation of residues that mediate binding of the substrate, ATP, and the transfer of the phosphate group, thus converting a kinase from the inactive to the active conformation [17]. Trans-phosphorylation from an upstream kinase as part of a multi-step signaling cascade [18], trans-autophosphorylation by an active kinase [19], and cis-autophosphorylation are the three main pathways for activation loop phosphorylation [20]. Despite the mechanism of autophosphorylation in the activation loop is not well understood, many vital kinases are regulated in this way [19,20,21,22,23,24]. The tyrosine residue phosphorylation in the activation loop of GSK3 (glycogen synthase kinase-3) was confirmed to be an autophosphorylation event [24]. Ser/Thr kinase HIPK2 (homeodomain-interacting protein kinase 2) was heavily modified by autophosphorylation at Tyr354 and Ser357 in the activation loop to regulate gene expression and cell proliferation [20]. Thus, autophosphorylation plays a vital role in kinase activity regulation.

Though it is distantly related to other Ser/Thr kinase family, an atypical activated loop ASCH (activation segment C terminal helix) was also found in the catalytic domain of STK16, suggesting it as a new member of the NAK family [9,25]. Moreover, suggested by its crystal structure, STK16 has a well-ordered catalytic conformation of the activation loop without dependence on exogenous phosphorylation induction and is constitutively active. Further mass spectrometry studies showed autophosphorylation is the main activation pattern. STK16 is capable of autophosphorylate at Thr185, Ser197, and Tyr198 in the activation segment. In the structure of unphosphorylated STK16, Ser197 and Tyr198 are buried in the hydrophobic cleft, resulting in a significant difference in the autophosphorylation process between the three sites. Phosphorylation at the Thr185 site is the fastest, requiring about 1 h, followed by Ser197 (3 h) and Tyr198 phosphorylation (18 h) [9]. However, whether these sites participate in the regulation of STK16 kinase activity, localization, and functions *in vivo* is still unknown.

In this paper, by site-directed mutagenesis on the three autophosphorylation sites of STK16, we found that the Tyr198 is not only critical for the Golgi apparatus and cell membrane localization of STK16 in cells, but also important for its kinase activity, Golgi apparatus structure and cell cycle progression.

## 2. Results

### 2.1. Subcellular Localization of STK16 in Mitosis

Since there are no STK16 antibodies that work for the detection of endogenous human STK16 in immunofluorescence, in our previous study, we used exogenously transfected GFP/FLAG-tagged human STK16 in HeLa cells and found that STK16 mainly associates with the Golgi structure in both mitosis and interphase [10]. To further unveil the detailed subcellular localization of STK16 in mitosis, we co-stained GFP/FLAG-tagged human STK16 and Giantin, an integral membrane protein that resides in the Golgi complex. As the Golgi apparatus undergoes fragmentation and reorganization during mitosis [26,27,28], Giantin is often used to monitor the structural change of Golgi [10,28,29]. In addition, we also stained M6PR and TGN46 respectively, two other markers of Golgi apparatus, and the results were shown in Appendix A.

During interphase, STK16 was mainly localized on the Golgi complex and cell membrane (Figure 1A), but a part of STK16 dissociates from the fragmented Golgi apparatus in mitosis (Figure 1B–H). Especially after cells entering metaphase (Figure 1D,E) and anaphase (Figure 1F), when the Golgi apparatus has been transformed into Golgi haze, most of the STK16 localized on Golgi apparatus was distributed into the cytoplasm, leaving only a small amount co-localizes with the Golgi haze. During telophase (Figure 1G) and cytokinesis (Figure 1H), STK16 was repositioned on the reassembled Golgi apparatus. In addition, the cell membrane localization of STK16 is also reduced from prometaphase to metaphase. These results suggest that although STK16 is mainly localized on the Golgi apparatus and the cell membrane in interphase, a portion of STK16 was dissociated from these locations and enters cytoplasm in mitosis.

### 2.2. The Maintenance of the Golgi Apparatus Structure Depends on STK16-Tyr198 Phosphorylation

To investigate the effect of the three autophosphorylation sites of STK16, Threonine185, Serine197, and Tyrosine198 were substituted with glutamate (mimic phosphorylated state) or alanine (mimic unphosphorylated state) by site-directed mutagenesis. Primers involved in construction of mutants are shown in Appendix A. We constructed expression vectors containing the human STK16 mutant cDNA with GFP and FLAG fused at STK16 C-terminus and transfected them into HeLa cells to establish HeLa-STK16 mutant-GFP-FLAG stable cell lines, including T185E, T185A, S197E, S197A, Y198E, Y198A, 2E (S197E-S198E), 2A(S197A-Y198A), 3E (T185E-S197E-Y198E), and 3A(T185A-S197A-Y198A). These mutants have shown normal growth rate and morphology. Next, we used immunofluorescence to examine their effects on subcellular localization of STK16 (Figure 2). Compared with HeLa-STK16 WT-GFP-FLAG stable cell lines (Figure 2B), the single site mutation of T185E, T185A, S197E, and S197A had no effect on the subcellular localization of STK16 and the Golgi apparatus structure (Figure 2C–F). However, both Y198E and Y198A result in the dissociation of STK16 from the Golgi apparatus and cell membrane (Figure 2G,H). Meanwhile, the Golgi apparatus was also disassembled. Moreover, it is interesting that the impact of Y198E seems greater than Y198A, as Y198E completely destroyed the Golgi apparatus localization of STK16 and caused severe fragmentation of the Golgi apparatus.

It is obvious that in all single, double, and triple mutants that included Y198E, STK16 completely lost its Golgi localization and caused the Golgi apparatus disassembly. It should be noted that there was no phenotypical difference among these mutants, suggesting that phosphorylation of Tyr198 alone was sufficient to dissociate STK16 from the Golgi apparatus and cause the Golgi disassembly.

We also found that the Tyr198 mutation affected the cell membrane localization of STK16, which also caused the absence of cell membrane localization of STK16 in double and triple mutants. The fact that there seems to be no difference between Y198E and Y198A indicates that the cell membrane localization of STK16 might depend on the dynamic auto-phosphorylation balance at Tyr198, which means that both phosphorylation and de-phosphorylation will impair the cell membrane localization of STK16.

### 2.3. The Localization of STK16 to the Golgi Apparatus is Dependent on its Kinase Activity

According to Figure 2, we know that Tyr198 mutations significantly affect the subcellular localization of STK16. However, it is not clear whether they are directly related to its kinase activity. It has been shown that STK16-IN-1, a specific STK16 kinase inhibitor, efficiently inhibits the kinase activity of STK16 both *in vitro* and *in vivo* [10,12]. Therefore, we combined STK16-IN-1 to examine the relationship between the subcellular localization of STK16 and kinase activity. As shown in Figure 3A, although it seemed not to be obvious in HeLa-STK16-WT-GFP-FLAG cells due to its dominant Golgi localization, when the kinase activity of STK16 Y198E-GFP-FLAG and STK16 Y198A-GFP-FLAG was inhibited by STK16-IN-1, STK16 was relocalized to the Golgi apparatus to a great extent, suggesting that the kinase inactive form of STK16 is localized on the Golgi apparatus. This is consistent with the results in Figure 2, although Y198A alone reduces the Golgi location of STK16, S197 and T185 mutations restore its Golgi localization, which indicates that the dephosphorylation of S197 and T185 is a positive signal for STK16 to localize to Golgi. Moreover, the structure of the Golgi apparatus is closely related to the localization of STK16 (Figure 2 and Figure 3A).

To rule out the off-target possibility of STK16-IN-1, we constructed the STK16-Y198E/Y198A-F100C-GFP-FLAG stable cell lines. Phenylalanine100 is substituted with cysteine, causing STK16 to lose its affinity with STK16-IN-1 [12]. As shown in Figure 3B, when STK16-IN-1 lost its inhibition on STK16-Y198E-F100C and STK16-Y198A-F100C, STK16-Y198E and STTK16-Y198A no longer returned to the Golgi apparatus, which confirms the specificity of STK16-IN-1 induced Golgi localization of STK16.

### 2.4. Phosphorylation at Tyr198 Affects the Localization of STK16 on the Cell Membrane

To evaluate the impact of phosphorylation on the cell membrane localization of STK16, we separated the cell membrane and cytoplasmic components of HeLa-GFP-FLAG (vector control), WT and mutants of HeLa-STK16 -GFP-FLAG cell lines and quantified the relative amount of STK16 in each fraction by Western blot analysis. GAPDH (used as a cytoplasmic maker), Histone H3 (used as a nuclear marker), and EGFR (used as a cell membrane maker) were used to confirm the separation efficiency. As shown in Figure 4A, compared with WT, the concentrations of single phosphorylation site mutants T185E, T185A, S197E, and S197A did not affect STK16 on cell membrane. However, for mutants containing Tyr198 mutation (Y198E, Y198A, 2E, 2A, and 3E), the proportions of STK16 on the cell membranes were all significantly reduced, except for 3A (Figure 4A,B). STK16-3A largely restored the cell membrane localization, which was similar to its Golgi localization. Quantitatively, in HeLa-STK16 Y198A-GFP-FLAG cells, the relative amount of STK16 on cell membrane was decreased by more than 50% (Figure 4C). These results suggested that Tyr198 is the key residue to the cell membrane localization of STK16.

### 2.5. STK16’s Kinase Activity is Differentially Regulated by its Phosphorylation on Ser197/Tyr198

4EBP1 is a substrate of STK16 [4]. To test whether the kinase activity of STK16 is dependent on its three auto-phosphorylation sites, we firstly expressed and purified STK16 WT, various STK16 mutants, and 4EBP1 proteins and conducted *in vitro* phosphorylation assays. The Thr185 and S197 mutants did not alter its kinase activity, while the mutation of Tyr198 significantly reduced the phosphorylation level of 4EBP1 by STK16 (Figure 5A). Meanwhile, we detected the autophosphorylation activity of STK16 WT and STK16 mutant proteins with a general Tyr/Ser/Thr phosphorylation antibody (Figure 5B). It seems both 197 and 198 mutations could reduce STK16 autophosphorylation on Ser/Thr, while only 198 mutations reduced its phosphorylation on 4EBP1. Moreover, it is either only one Tyr phosphorylation site on STK16, or the 198 mutations completely abolished its autophosphorylation on all Tyr residues.

### 2.6. The Effect of STK16 on Cell Cycle is Also Dependent on Tyr198 Phosphorylation

We previously found that STK16 RNAi or kinase activity inhibition by STK16-IN-1 could inhibit G2/M transition and delay mitotic progression [10]. By a synchronization experiment, here we found that overexpression of STK16-WT promoted cell transition from S to G2/M and delayed the mitosis exit (Figure 6A). However, it is very interesting that although overexpression of the Y198E caused a prolonged G2/M phase (Appendix A), overexpression of the Y198A mutant had an indistinguishable phenotype from the vector control (Figure 6A), which indicates that the mutation of Y198A abolished the cell cycle regulation function of STK16. Moreover, Western blot using phospho-Histone H3 (Thr3), phospho-Histone H3 (Ser10), and cyclin B1 as mitotic and G2/M markers, also confirmed that Y198A mutation abolished the cell cycle regulation function of STK16 (Figure 6B,C).

## 3. Discussion

It is already known that phosphorylation in the activation loop is required for activity regulation of many kinases [30,31], but the phospho-regulation of STK16 has never been reported. Among the three autophosphorylation sites of STK16 in the activation loop, Thr185 is located in ASCH, while Ser197 and Tyr198 are located in the p+1 ring following the atypical activation loop ASCH [9], which suggest that the kinase activity of STK16 may be regulated by these three sites. Although our *in vitro* phosphorylation assays confirmed that all these three sites were critical for STK16’s kinase activity, Tyr198 was actually the most critical residue for its kinase activity, Golgi localization as well as cell cycle regulation functions, although STK16 is a Ser/Thr kinase. According to recent studies, it seems not uncommon that the autophosphorylation of Tyr in the activation loop could control its Ser/Thr kinase activity. For example, the dual-specificity tyrosine-phosphorylation-regulated kinases (DYRKs) are Ser/Thr kinases that only autophosphorylate the second tyrosine of the activation loop YxY motif during protein translation. Once Tyr is phosphorylated, the DYRK activation segment is stabilized [32]. Another report in regard to the Ser/Thr kinase GSK3β found the autophosphorylation at its Tyr216 residue constitutively activated this kinase and determined the kinase activity [33]. Thus, autophosphorylation of Tyr198 in STK16 may be essential for its activation loop stabilization and constitutively activation.

Besides STK16, AAK1 (adaptor-associated kinase 1), BIKE/BMP2K (BMP-2-inducible kinase), and GAK (cyclin G-associated kinase) are the other three members of the NAK kinase family [25]. All of them were initially reported to participate in cellular processes such as cell signaling and protein secretion and sorting, but they were later found to be related to cell division through interacting with mitotic related proteins [10,34,35,36], and affecting mitotic centrosome maturation [37,38] or microtubules outgrowth [39], etc. Moreover, the effect of the GAKs on cell division is dependent on their kinase activity by phospho-regulation. For example, Y412 and Y1149 are the two phosphorylation residues playing significant roles in cell cycle progression [37]. For STK16, when the kinase activity was inhibited by STK16-IN-1, it had no effect on the cell cycle [10]. Consistent with our previous conclusion, here as shown in Figure 6, by double thymidine synchronizing HeLa cells overexpressed STK16 and STK16 mutants, we found that the overexpression of STK16 wild type promotes cell transition from S to G2/M and delayed the mitosis exit, but not the Y198A mutant. That is to say, STK16 is dependent on phosphorylation at Tyr198 to maintain its activity to participate in cell cycle regulation.

The Golgi apparatus is the core component of the endomembrane system in eukaryotes, playing a crucial role in the cellular activities, such as post-translational modification and trafficking of protein, lipid biosynthesis, etc. [40]. Moreover, highly dynamic and precisely regulated mitotic progression and orderly fragmentation and reorganization of the Golgi apparatus are interdependent [41,42]. In our study, as a Golgi-resident protein, STK16 was primarily localized on the Golgi apparatus and cell membrane during interphase. However, when cells entered mitosis, STK16 left the Golgi apparatus and entered the cytoplasm until telophase. This localization change might be related to the mitotic progression. In fact, it has been reported that multiple Golgi resident proteins are dissociated from the Golgi apparatus during mitosis and participate in cell cycle regulation. For instance, Nir2 is phosphorylated by CDK1 in mitosis, dissociates from the Golgi, and forms Nir2-Plk1 interaction, which targets the cleavage furrow and midbody to regulate cytokinesis completion [43]. ACBD3 binds to the Golgi protein Giantin during interphase, and releases from the Golgi apparatus during mitosis, when it regulates the cell cycle and determines the cell fate by affecting the numb signaling pathway [44]. Clathrin dissociates from the Golgi during mitosis and concentrates at the spindle apparatus, stabilizing fibers of the mitotic spindle to aid chromosome congression [45]. These suggest that the Golgi resident proteins regulate the progression of cell division by altering its localization during mitosis or interacting with other proteins in the cell.

Our results suggested that activation loop autophosphorylation of STK16 could regulate its localization and signal transmission. Specifically, the Tyr198 residue of STK16 was critical for its kinase activity, Golgi and cell membrane localization, as well as its functions in regulating Golgi apparatus structure and cell cycle progression. Further studies are necessary to characterize its potential upstream kinases and downstream substrates in cells, especially in cell cycle regulation, which are not only important for understanding the function of STK16, but also for the intertwined connection between the Golgi and cell cycle regulation.

## 4. Materials and Methods

### 4.1. Cell Culture and Establishment of Stable Cell Lines

HeLa cells were grown in DMEM without L-glutamine (Corning Life Sciences, Manassas, VA, USA), supplemented with 10% (*v*/*v*) fetal bovine serum (Clark, Richmond, VA, USA), 100 units/mL of penicillin-100 µg/mL of streptomycin (Hyclone, Logan, UT, USA), 2 mM glutamax (Gibco, Carlsbad, CA, USA), 5% CO_2_, at 37 °C in incubators. Human STK16 cDNA was cloned into a pMSCV-puro vector with GFP and 3 × FLAG tags fused at the C-terminal to form STK16-GFP-FLAG wild type plasmid. STK16-T185E-GFP-FLAG and STK16-T185A-GFP-FLAG plasmids were point mutated by substituting nucleotides ACC for GAA and GCG at the 553–555th respectively, STK16-S197E-GFP-FLAG and STK16-S197A-GFP-FLAG plasmids were point mutated by substituting nucleotides TCC for GAA and GCC at the 588–591th respectively, and STK16-Y198E-GFP-FLAG and STK16-Y198A-GFP-FLAG were point mutated by substituting nucleotides TAC for GAA and GCC at the 592–594th respectively. In addition, STK16-S197EY198E (2E)/S197AY198A (2A)/T185ES197EY198E (3E)/T185AS197AY198A (3A) plasmids were generated similarly. HeLa-STK16-GFP-FLAG cell lines were established by the retrovirus system. Retrovirus were packaged by transfecting these plasmids cloned into indicated genes with two helper plasmids into 293 T cells using Fugene 6 (Promega, Madison, WI, USA. The supernatants containing viruses were harvested after 48 h of incubation and infected the HeLa cells. Stable cell lines were screened by puromycin (Selleck, Houston, TX, USA) and maintained in a medium containing 1 μg/mL puromycin.

### 4.2. Reagents

The anti-GFP (ab290), p-Ser/Thr (ab17464), Histone H3 (ab198757), and Giantin (ab37266) antibodies were acquired from Abcam (Cambridge, Britain). The anti-phospho-H3-S10 (3377) and anti-phospho-H3-T3 (9714), EGFR (4267), CyclinB1 (12231), 4EBP1 (53H11), p-4EBP1 (2855), p-Tyr (9411), p-Thr (9386) antibodies, and the HRP-linked anti-rabbit and anti-mouse IgG antibody were from Cell signaling technology (MA, USA). The anti-GAPDH and β-tubulin antibodies were from Beijing TransGen Biotech (Beijing, China). The secondary antibodies and anti-fade Prolong Gold with DAPI were from Invitrogen (Carlsbad, CA, USA). The anti-STK16 and anti-FLAG M2 monoclonal antibody (F3165) were from Sigma. GlutaMAX supplement was from Gibco. Puromycin dihydrochloride was from Selleck. All PCR primers were synthesized and genes were sequenced by Sangon Biotech (Shanghai, China). A Membrane and Cytosol Protein Extraction Kit was purchased from Beyotime (Shanghai, China).

### 4.3. Construction of Prokaryotic Expression Vector and Protein Purification

Human STK16 cDNA (GenBank accession number gi4505837) was subcloned into a pGEX-4T-1 vector, expressed in *Escherichia. coli* (BL21 [DE3]) cells and purified by glutathione affinity resin. The GST-STK16 WT protein obtained was further cleaved off the GST tag by thrombin protein. GST-STK16 mutants were point mutated by replacing the corresponding residues and cloned into the same vector, followed by purification with slight modifications. Human 4EBP1 cDNA was cloned into a pET-28a vector, expressed in *E. coli* (BL21 [DE3]) cells and purified by Ni-NTA His Bind Resin.

### 4.4. Immunofluorescence

HeLa-GFP-FLAG, HeLa-STK16 WT-GFP-FLAG, and HeLa-STK16 mutant-GFP-FLAG cells were grown on coverslips and treated with the drugs for indicated time points. Cells were washed once with PBS to remove residual culture media and fixed by −20 °C methanol for 5 min. Then coverslips were washed with TBS-Tx (TBS supplemented with 0.1% Triton X-100) and blocked by AbDil-Tx (TBS-Tx supplemented with 2% BSA and 0.05% sodium azide) at room temperature for 30 min, and stained with primary antibodies (anti-GFP and anti-Giantin). The secondary fluorescently conjugated antibodies were probed at room temperature for 1 h. After removing unbound secondary antibodies by TBS-Tx, coverslips were mounted in anti-fade Prolong Gold with DAPI. Images were taken using a Leica MI4000B fluorescent microscope. All experiments were repeated at least three times and representative micrographs are shown in the figures.

### 4.5. Western Blots

Cells were washed once with PBS to remove the residual medium, then were lysed by M-PER^TM^ Mammalian Protein Extraction Reagent (Thermo Scientific, Waltham, MA, USA) buffer supplemented with a protease and phosphatase inhibitor cocktail (Roche, Basel, Switzerland) at 4 °C for 30 min. The cell lysate was completely collected and thermally denatured at 95 °C for 8 min with 5 × SDS loading buffer. The sample was electrophoresed on 12% SDS-PAGE gels in Bio-Rad Mini-PROTEAN Tetra Cell, which was then transferred onto the PVDF (polyvinylidene fluoride) membranes (Merck, Whitehouse Station, NJ, USA) by Thermo Scientific Owl VEP-2. The PVDF membrane was blocked with 5% non-fat dried milk at room temperature for 1 h, and then incubated with corresponding primary and HRP-conjugated secondary antibodies. Images were visualized by using enhanced chemiluminescence Kits (Millipore or Thermo Scientific) and Tanon Fine-do X6. ImageJ software (NIH, Bethesda, Maryland, USA) was used to quantify the relative protein levels.

### 4.6. In Vitro Phosphorylation Assay

Kinase assays were performed for 30 min at 37 °C in a final volume of 40 μL consisting of the kinase buffer (50 mM Tris, pH 7.4, 20 mM MgCl_2_, 2.5 mM DTT, and 20 μM ATP), 300 nM STK16 protein as kinase and 10 μM 4EBP1 as substrate. Reactions were stopped by the addition of 5ХSDS loading buffer and boiled for 8 min. Samples were subsequently analyzed by SDS-PAGE and Western blotting.

### 4.7. Cell Synchronization

HeLa-GFP-FLAG, HeLa-STK16 WT-GFP-FLAG, and HeLa-STK16 mutant-GFP-FLAG cells were seeded at 30% confluence in 35 mm dishes for one day before double-thymidine block assay. Cells were firstly blocked with 2.5 mM thymidine in DMEM for 16 h. After being washed three times with warm PBS to remove thymidine, cells were released for 8 h with fresh DMEM medium. Then DMEM containing 2.5 mM thymidine was added again for another 16 h to retain cells at G1/S transition. After being washed three times with warm PBS, cells proceed sequentially from the G1/S transition. “0” indicated the beginning of double thymidine release. Cells were collected at indicated times after being released for SDS-PAGE and Western Blot or fixed for FACS.

### 4.8. Cell Cycle Analysis by FACs

HeLa-GFP-FLAG, HeLa-STK16 WT-GFP-FLAG, HeLa-STK16 Y198E-GFP-FLAG, and HeLa-STK16 Y198A-GFP-FLAG cells were synchronized to G1/S border by double-thymidine, and released into different stages to be trypsinized and collected, then thoroughly cleaned before being fixed by 70% ice-cold ethanol overnight at −20 °C. Cells were washed to remove ethanol by PBS and stained with Propidium Iodide solution (BD Pharmingen, San Diego, CA, USA) as the manufacturer’s protocol indicated. Samples were detected by Backman flow cytometry and data were analyzed by ModFit LT software (Verity Software House, LA, CA, USA).

## Figures and Tables

**Figure 1 ijms-20-04852-f001:**
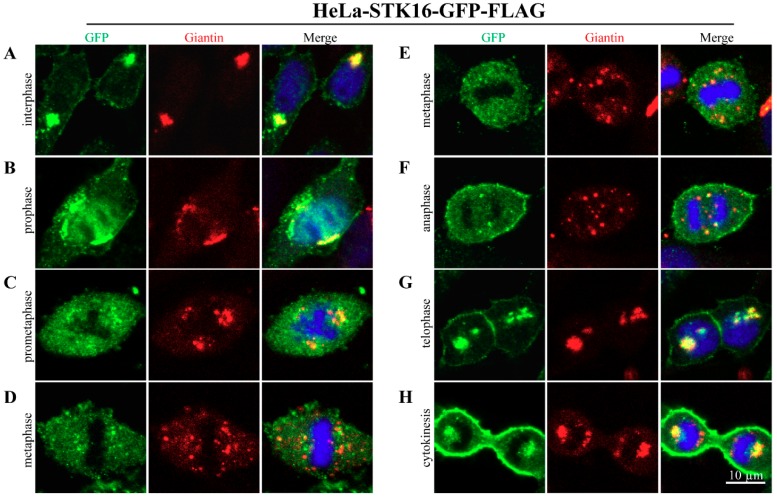
The subcellular localization of STK16 throughout the cell cycle. HeLa cells stably expressing STK16-GFP-FLAG were fixed and then subjected to the anti-GFP antibody and anti-Giantin antibody staining. GFP staining of STK16 is shown in green, Giantin staining is shown in red, and DAPI staining of the nucleus is shown in blue. (**A**) Interphase. (**B**) Prophase. (**C**) Prometaphase. (**D**,**E**) Metaphase. (**F**) Anaphase. (**G**) Telophase. (**H**) Cytokinesis. Experiments were repeated at least three times and representative results are shown. Scale bar, 10 μm.

**Figure 2 ijms-20-04852-f002:**
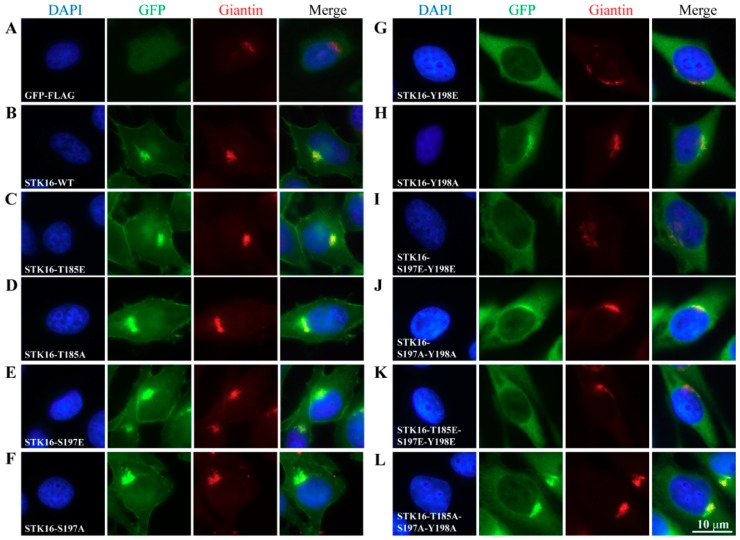
Effects of three autophosphorylation sites of STK16 on its subcellular localization. HeLa cells stably transfected with (**A**) GFP-FLAG, as well as GFP-FLAG tagged (**B**) STK16-WT, (**C**) STK16-T185E, (**D**) STK16-T185A, (**E**) STK16-S197E, (**F**) STK16-S197A, (**G**) STK16-Y198E, (**H**) STK16-Y198A, (**I**) STK16-2E (S197E and Y198E), (**J**) STK16-2A (S197A and Y198A), (**K**) STK16-3E (T185E, S197E, and Y198E), and (**L**) STK16-3A (T185A, S197A, and Y198A) were fixed and stained for GFP (green) and Giantin (red). DAPI staining of the nucleus is shown in blue. Experiments were repeated at least three times and representative results are shown. Scale bar, 10 μm.

**Figure 3 ijms-20-04852-f003:**
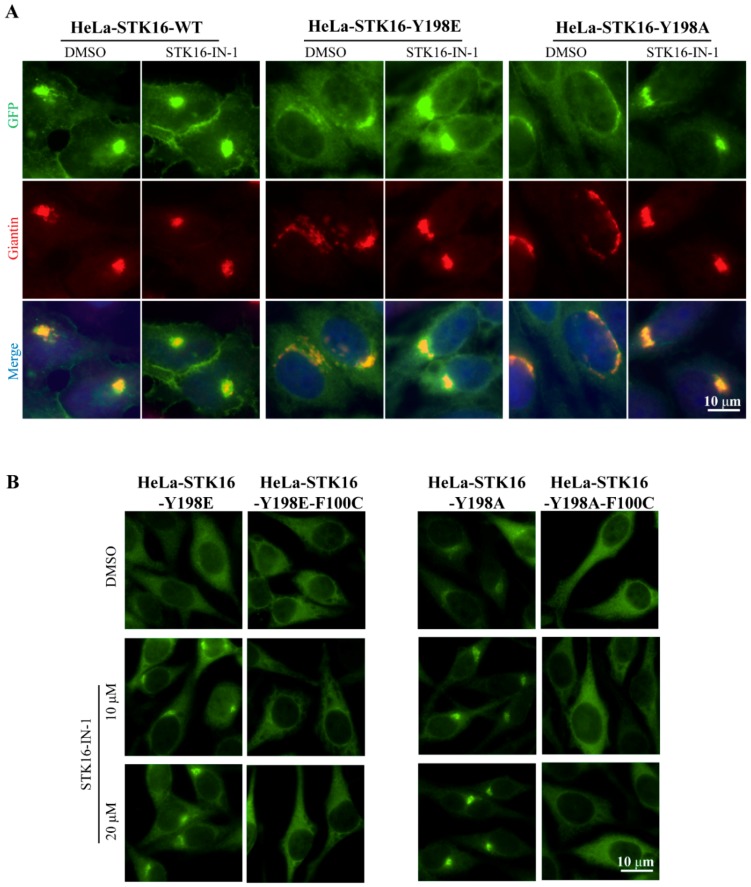
The kinase activity and Tyr198 phosphorylation of STK16 determines the localization of STK16. (**A**) HeLa-STK16 WT-GFP-FLAG, HeLa-STK16 Y198E-GFP-FLAG, and HeLa-STK16 Y198A-GFP-FLAG overexpression cell lines grown on coverslips were treated with DMSO or 10 μM STK16 IN-1 for 8 h. Immunofluorescence used anti-GFP (green) and anti-Giantin (red) antibodies to analyze the localization of STK16. (**B**) HeLa cells stably expressing STK16 Y198E-GFP-FLAG and HeLa-STK16 Y198E-F100C-GFP-FLAG were treated with DMSO or 10 μM, and 20 μM STK16-IN-1 for 8 h. Experiments were repeated at least three times and representative results are shown. Scale bar, 10 μm.

**Figure 4 ijms-20-04852-f004:**
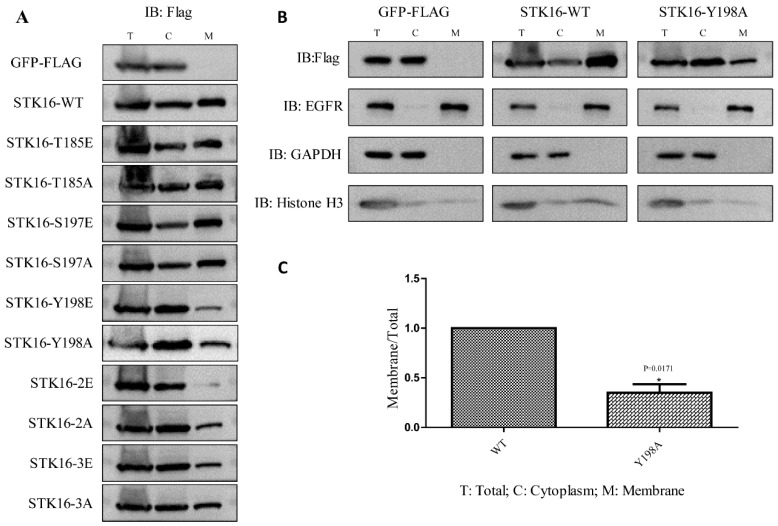
Phosphorylation at Tyr198 affects the localization of STK16 in the cell membrane. (**A**,**B**) HeLa cells stably expressing GFP-FLAG (vector control), STK16 WT-GFP-FLAG and STK16 mutants-GFP-FLAG were grown in 35 mm dishes with 90% confluence, and then fractionated into the cytoplasm (C) and cell membrane (M). (**A**) The amount of wild type and mutant STK16 in both components of each cell line were detected by Western blot. (**B**) Markers were used to verify fractionation efficiency. (**C**) Quantification of STK16 concentrations on the cell membrane and cytoplasm in HeLa-STK16 WT-GFP-FLAG and HeLa-STK16 Y198A-GFP-FLAG cells were shown. The data were analyzed using Student’s *t*-test. Data show mean ± SEM, *n* = 3.

**Figure 5 ijms-20-04852-f005:**
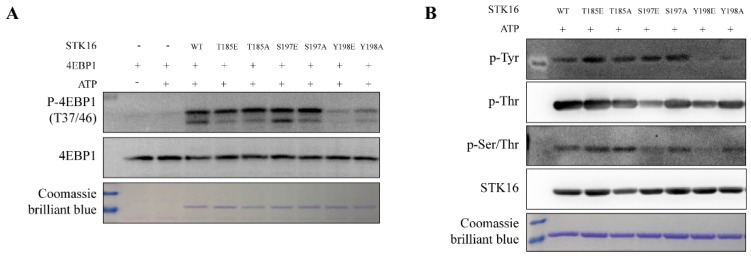
Tyr198 phosphorylation is important for the kinase activity of STK16. (**A**) *In vitro*, the mutation of Tyr198 significantly inhibited the phosphorylation of 4EBP1 by STK16. 300 nM STK16 or STK16 mutant proteins were incubated with 10 μM 4EBP1 protein at 37 °C for 30 min in kinase buffer. The 4EBP1 and phospho-4EBP1 (T37/46) antibodies were used in Western blots. (**B**) The Tyr198 mutation reduced the Tyr kinase activity of STK16. 1 μM STK16 or STK16 mutant proteins were incubated at 37 °C for 30 min in kinase buffer. p-Tyr, p-Thr, p-Ser/Thr, and STK16 antibodies were detected by Western blots. Coomassie brilliant blue stained STK16 and STK16 mutant proteins.

**Figure 6 ijms-20-04852-f006:**
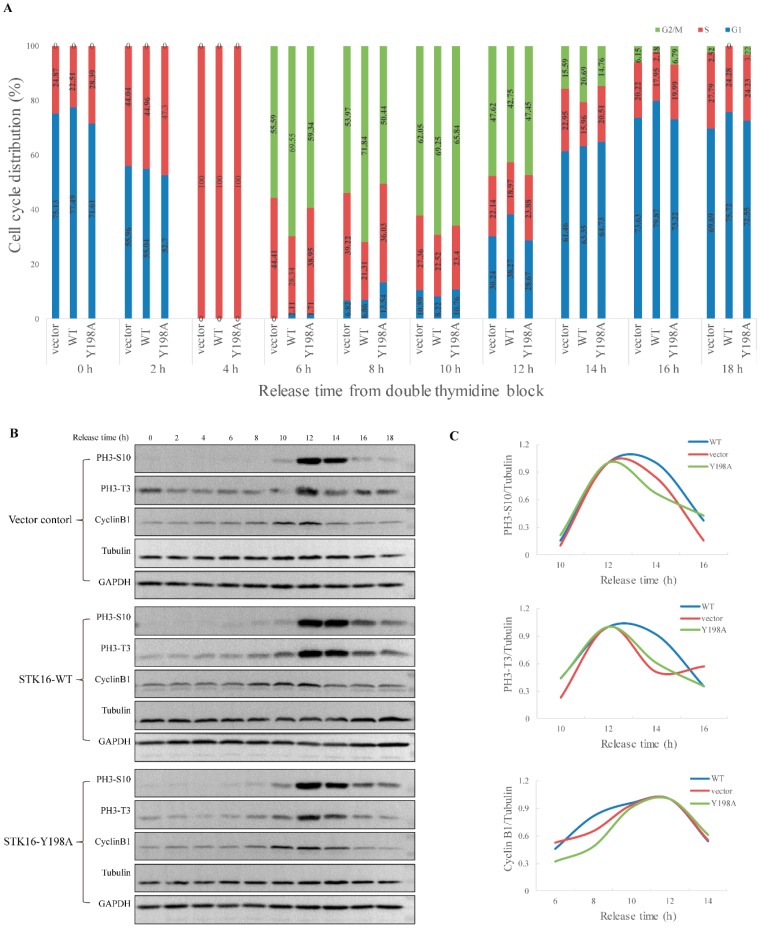
Tyr198 is critical for STK16 to regulate cell cycle. HeLa-GFP-FLAG (vector control), HeLa-STK16 WT-GFP-FLAG, and HeLa-STK16 Y198A-GFP-FLAG cell lines were synchronized by double thymidine release, and the cell cycle was detected by fluorescence-activated cell sorting (FACS) (**A**) and Western blot (**B**) at the times indicated. (**C**) Phospho-Histone H3 (Thr3), phospho-Histone H3 (Ser10), and cyclin B1 were normalized to tubulin. Phospho-Histone H3 (Thr3) and phospho-Histone H3 (Ser10) were used as the mitotic and G2/M makers in Western blots.

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
