# Peer review of "Tyr198 is the Essential Autophosphorylation Site for STK16 Localization and Kinase Activity"

_ijms, 2019, doi:10.3390/ijms20194852_

Round 1

Reviewer 1 Report

I find your paper well written, figures (of very high quality) are nicely explained, the results are clearly presented, discussion is interesting.

However, I have 2 main questions:

Why you did not make any comment on the effect of mutation Y198E on the cell cycle?

Why did not you make any comment on "overall" phenotype of the mutated HeLa cells? Was there any change in general physiology of mutated cell as compared to WT? Are they more sensitive to some cytostatic drugs?

It would be great if you could make any comment or if possible to do some experiments.

Reviewer 2 Report

The authors reported that Tyr198 is an essential autophosphorylation site for STK16 localization to the plasma membrane using immunofluorescence studies and Western blot analysis.
Both ICC and WB data are clear and compelling. The authors also reported that kinase activity was affected by the Tyr198 mutation.
This paper is well written and represents an important discovery.

Minor point
# 1. Figure 3A. It is important to display the merge image.

Reviewer 3 Report

The manuscript by Wang and colleagues describes shows that STK16 autophosphorylation at Tyr198 is essential for the kinase localization into the Golgi, maintenance of the Golgi structure as well as progression through the cell cycle. While the findings of the authors are of interest, this is a merely descriptive manuscript, relying mainly on a single technique: fluorescence. 

Several issues arose:

How does STK16 control golgi structure? What is the level of STK16 overexpression compared with endogenous protein? Have the authors conducted in vivo microscopy, in addition to immunofluorescence, to track STK16 subcellular localization? Why phospho-mimetic and non-phosphorylatable STK16 mutants behave the same? Is it likely that mutation into E is not a good phospho-mimetic. The author's experiment shown in Figure 5B suggest that S197 and T185 might not be strong autophosphorylation sites as there is no reduction on the pThr nor pSer/pThr immunoblots. Have other sites been identified to explain this finding? Figure 6A: could the authors provide actual % of cells in each phase (i.e. show numbers) and stastistical test?

Author Response

We appreciate that the reviewers have provided very constructive comments and suggestions, which are very important for us to improve our manuscript. We have used track change to highlight the major changes in the revised manuscript. The point-by-point responses to the reviewers are as follows.

Review #3

The manuscript by Wang and colleagues describes shows that STK16 autophosphorylation at Tyr198 is essential for the kinase localization into the Golgi, maintenance of the Golgi structure as well as progression through the cell cycle. While the findings of the authors are of interest, this is a merely descriptive manuscript, relying mainly on a single technique: fluorescence. Several issues arose:

How does STK16 control Golgi structure?

------------Sorry that we should have explained it clearly in our manuscript. In our group's previous work (Liu, et al., STK16 regulates actin dynamics to control Golgi organization and cell cycle. Sci Rep-Uk 2017. https://www.ncbi.nlm.nih.gov/pmc/articles/PMC5353726/), we have revealed STK16 as a novel actin binding protein that resides in the Golgi, which directly regulates actin dynamics to control Golgi structure.

What is the level of STK16 overexpression compared with endogenous protein?

------------We wanted to address this question ourselves too. However, since STK16-related studies are very limited, there are no good antibodies for STK16 that work well. We have bought a few commercially available antibodies, such as #SAB1406692 (Sigma-Aldrich) and ab37975 (Abcam) but found that most of them only work for purified STK16 proteins, but they either do not recognize endogenous STK16, or overexpressed STK16. We are in the process of making a customized STK16 antibody ourselves, which will need a few months. We hope that we are lucky enough to get an antibody that can recognize both endogenous STK16 and overexpressed STK16.

Have the authors conducted in vivo microscopy, in addition to immunofluorescence, to track STK16 subcellular localization?

------------This is a really good suggestion. But we are very sorry that we currently do not have a microscope for live imaging. We hope to address this question in the future.

Why phospho-mimetic and non-phosphorylatable STK16 mutants behave the same? Is it likely that mutation into E is not a good phospho-mimetic. The author's experiment shown in Figure 5B suggest that S197 and T185 might not be strong autophosphorylation sites as there is no reduction on the pThr nor pSer/pThr immunoblots. Have other sites been identified to explain this finding?

------------This is a really good question which we were actually confused for a long time ourselves. We originally also thought that the E mutation was not a good phospho-mimetic. However, we noticed that there were some differences between 198A and 198E. For example,

In figure 2, we can see that STK16 Y198E is completely dispersed into the cytoplasm, while STK16 Y198A has partial aggregation, and the difference is more obvious between STK16 3E and STK16 3A. Here is our hypothesis: since STK16 regulates actin dynamics in a kinase activity-dependent way (Liu et al, Scientific Reports.  2017, https://www.ncbi.nlm.nih.gov/pmc/articles/PMC5353726/, Figure 4), it is possible that both phospho-mimetic and non-phosphorylatable STK16 mutants significantly disrupt the balance of STK16 phosphorylation, both resulted in STK16 localization abnormality and cell cycle disruption. Besides T185, S197 and Y198, no other phosphorylation sites of STK16 have been identified so far.

Figure 6A: could the authors provide actual % of cells in each phase (i.e. show numbers) and statistical test? 

------------Thanks for pointing this out. We have now provided this information in the new Figure 6A.

Round 2

Reviewer 3 Report

I thank the authors for having addressed all of my concerns.